# Cerebellar climbing fibers multiplex movement and reward signals during a voluntary movement task in mice

Koji Ikezoe [1,5✉], Naoki Hidaka[1,2,5], Satoshi Manita[1], Masayoshi Murakami[1], Shinichiro Tsutsumi[2,3], Yoshikazu Isomura[4], Masanobu Kano [2✉] & Kazuo Kitamura [1,2✉]

Cerebellar climbing fibers convey sensorimotor information and their errors, which are used for motor control and learning. Furthermore, they represent reward-related information. Despite such functional diversity of climbing fiber signals, it is still unclear whether each climbing fiber conveys the information of single or multiple modalities and how the climbing fibers conveying different information are distributed over the cerebellar cortex. Here we perform two-photon calcium imaging from cerebellar Purkinje cells in mice engaged in a voluntary forelimb lever-pull task and demonstrate that climbing fiber responses in 68% of Purkinje cells can be explained by the combination of multiple behavioral variables such as lever movement, licking, and reward delivery. Neighboring Purkinje cells exhibit similar climbing fiber response properties, form functional clusters, and share noise fluctuations of responses. Taken together, individual climbing fibers convey behavioral information on multiplex variables and are spatially organized into the functional modules of the cerebellar cortex.

[1] Department of Neurophysiology, Faculty of Medicine, University of Yamanashi, Chuo, Yamanashi 409-3898, Japan. [2] Department of Neurophysiology, Graduate School of Medicine, The University of Tokyo, Tokyo 113-0033, Japan. [3] Laboratory for Multi-scale Biological Psychiatry, RIKEN Center for Brain Science, Wako, Saitama 351-0198, Japan. [4] Department of Physiology and Cell Biology, Graduate School of Medical and Dental Sciences, Tokyo Medical and Dental University, Tokyo 113-8510, Japan. [8] These authors contributed equally: Koji Ikezoe, Naoki Hidaka. ✉email: kikezoe@yamanashi.ac.jp; mkano-tky@m.u-tokyo.ac.jp; kitamurak@yamanashi.ac.jp

Climbing fibers (CFs), axons of the inferior olivary neurons of the medulla, innervate Purkinje cells (PCs), which are the exclusive output neurons of the cerebellar cortex[1,2]. Each PC receives strong excitatory inputs from a single CF in the adult cerebellum[1,3], and CF activation induces a characteristic burst of spikes in the PC soma, termed complex spike (CS)[1]. CSs encode sensorimotor errors during voluntary movements; thus, CF inputs can act as instructive signals for motor learning[4–6]. They also encode the movement itself, such as its kinematics[7–11] and touch positions[12]. Moreover, they encode non-motor signals, such as reward-related signals during classical/operant conditioning tasks[13–17], temporal-difference prediction errors in eyeblink conditioning[18], or decision errors in a decision-making task[19]. These studies suggest that CFs convey both motor and non-motor information[20]; nonetheless, there is limited evidence for the extent to which individual CFs encode multiple modalities of information during behavior.

The cerebellar cortex comprises a parasagittal modular architecture characterized by CF projections, termed zones[21]. Adjacent PCs form microzones[22,23], and project their axons to a specific subdivision of the cerebellar nuclei, which in turn projects to a subnucleus of the inferior olive that provides CFs to the PCs in the identical cluster[24]. The aforementioned tripartite closed-loop circuit is termed a microcomplex and is regarded as a functional unit of the cerebellum[25]. Consistently, the CSs of PCs are synchronized in narrow bands along the rostrocaudal axis[26,27], and the synchrony is enhanced during skilled movements[28]. Two-photon calcium imaging also revealed microzonal CS activity during sensory stimulation[29–32] and voluntary movements[14,33–38]. Furthermore, our group has shown that averaged activities of microzonal CSs encode movement variables and non-motor variables, such as auditory cues and reward outcomes during the go/no-go licking task[17]. However, little is known about how individual CFs encode motor and non-motor information within each microzone.

In this study, we performed two-photon calcium imaging in cerebellar PCs of mice during a self-initiated lever-pull task[39–42] to elucidate if and how CSs encode multiple behavioral variables of different modalities in single PCs. The task is suitable for tracking both motor activities (lever pulling and licking movements) and task-related non-motor information (outcome or reward). Using encoding model analysis, we demonstrated that 68% of PCs encode multiple behavioral variables in their CS activity. These PCs were modulated by both lever movement and licking, and 23% of them were additionally modulated by reward delivery. Response properties were classified into eight types by using five behavioral variables, and PCs showing similar response properties were spatially clustered with sharp boundaries. Noise fluctuations of CS response in the same clusters were correlated more strongly than those in the different clusters, suggesting that each cluster corresponds to a microzone. Together, each PC in a substantial fraction of microzones receives multiplexed motor and reward information through a CF during behavior.

## Results

**Two-photon calcium imaging of PC dendrites in the cerebellum of mice engaged in a lever-pull task.** Mice were trained to perform a self-initiated lever-pull task (Fig. 1a and Supplementary Fig. 1). Briefly, head-restrained mice had to pull the lever with their left forelimb following a stationary period of 1 s at the resting position and hold the lever for 400 ms to obtain a water reward. Following six days of training (mean ± standard deviation: 5.6 ± 1.9 days, range, 4–10 days, 8 mice), the mouse achieved 400 ms hold-duration with the success frequency of 2 times/min. Subsequently, we conducted two-photon calcium imaging from

dendrites of PCs expressing a genetically encoded calcium indicator GCaMP6f (Fig. 1), wherein $Ca^{2+}$ signals have been demonstrated to reflect CSs of PCs[37].

During the imaging session, the mice sporadically and spontaneously pulled the lever and held it at the end of the movement range, followed by pushing it back (see Methods) to the resting position. The mice kept grabbing the lever in the task, even in the period of pushing the lever. The median frequency of the pull was 139 times per session of 550–600 s (Fig. 1b, e, range, 108–239 times, 8 mice, 9 imaging sessions). The mice obtained a water reward (4–8 μL) 380 ms following the 400-ms holding period (task success, Supplementary Fig. 1). The median number of successes in a session was 13 times (range, 4–50; Fig. 1b), suggesting that the mice were not fully trained. We examined

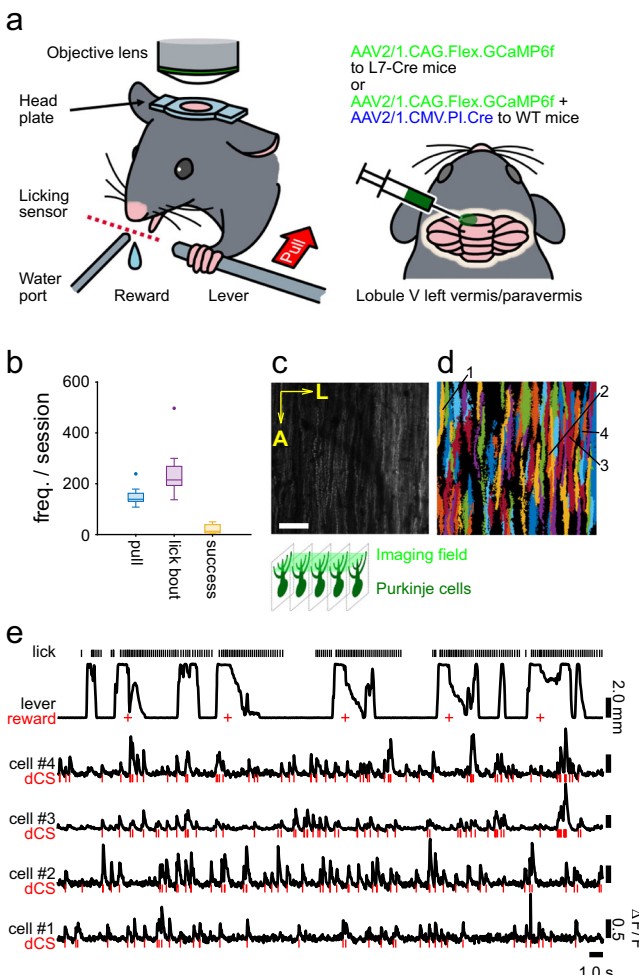

**Fig. 1 Two-photon calcium imaging of Purkinje cell (PC) dendrites during a self-initiated lever-pull task. a** A schematic of the experimental setup for two-photon imaging from the cerebellum of mice engaged in the behavioral task and animal preparation for calcium imaging. **b** The frequency of behavioral outputs in nine imaging sessions. Box-plot elements: center line, median; box limits, upper and lower quartiles; whiskers, maximum and minimum values within 1.5× interquartile range; points, outliers. **c** A fluorescence image of an imaging field. The dendrites of PCs appear in a striped pattern. A: anterior, L: lateral. Scale bar: 50 μm. **d** Extracted dendrites of the PCs from the image in **c**. **e** An example of mouse behavior and fluorescence signals of four dendrites. The cell numbers correspond to those in **d**. dCS: estimated spikes from fluorescence by deconvolution. Red bars and crosses represent the timings of dCS and reward delivery, respectively.

whether pull speed and holding duration changed during individual sessions. We split the pull events in half according to order in individual sessions, then compared the speed of pull and holding duration. We did not find the difference between halves of the speeds ($p = 0.11$–$0.98$, Wilcoxon's rank-sum test). Also, we did not find the difference between halves of the holding durations ($p = 0.21$–$0.96$, Wilcoxon's rank-sum test) except for an imaging site ($p = 0.0002$). At the population, we also did not find the difference between halves of the pull speed and of the holding durations (42 mm/s, 73 mm/s, 115 mm/s, and 42 mm/s, 80 mm/s, 114 mm/s; 1st, 2nd, 3rd quartile of pull speed in the 1st and 2nd halves, respectively; 90 ms, 265 ms, 436 ms, and 100 ms, 265 ms, 440 ms; 1st, 2nd, 3rd quartile of hold-duration in the 1st and 2nd halves, respectively). The mice licked the water reward and often began licking the water port before the lever pulls regardless of the task's success. Therefore, the number of lick bouts was considerably larger than that of the reward (median 215, range, 137–497, Fig. 1b, e). We did not present any cues to trigger lever movement and did not fully train mice; thus, the trajectory of lever movements displayed considerable variability in their speeds, magnitudes, intervals, and holding durations (Fig. 1e). We calculated the ratio of the third quartile to the first quartile of the speed of lever-pulls and the duration of lever-holding periods in the individual sessions. Mean ± s.e.m. was $2.4 \pm 0.03$ for the lever-pull speed and $5.7 \pm 0.3$ for the duration of a lever-holding period. These variabilities lowered the correlation among the behavioral variables, enabling us to examine the response properties of cells effectively.

During the task, we simultaneously imaged fluorescence signals in multiple dendrites of PCs located in the lobule V of the cerebellar vermis ipsilateral to the lever, which receives forelimb-related CF inputs[14,37,43] (Fig. 1a, c). They often displayed a transient increase in fluorescence (Fig. 1e), which was considered a proxy for CS occurrence[44]. We obtained fluorescence changes in individual dendrites using the Suite2P software[45] (Fig. 1d) and estimated the spike timing using the deconvolution technique[46] (deconvolved complex spikes (dCSs), Fig. 1e). The average dCS rate of PCs was $1.06 \pm 0.60$ Hz (517 cells), while dCS often showed short inter-spike intervals, consistent with previous reports on the mean rate and short inter-spike intervals of CSs[47–49]. Several dCSs appeared around behavioral events, namely, lever pulls, pushes, licking, and rewards, thereby suggesting that CSs in these cells encode information about the aforementioned behavioral events.

**Encoding model predicted the CSs of individual PCs during the lever-pull task.** To characterize the response properties of each PC to individual behavioral variables, we built encoding models with lever movement and other behavioral variables. An encoding model that can predict dCS activities during a task enables the description of the magnitude and timing of the cell's response to individual behavioral variables[50,51]. We applied a linear-nonlinear cascade model with a logistic function[8,52] (Fig. 2a). Five behavioral variables were selected as the input variables as follows: lever position, lever pull and push speeds, instantaneous lick rate, and water reward delivery (Supplementary Fig. 2a). We optimized the five linear response kernels for the individual behavioral parameters and an intercept to fit model outputs to 85% of the data (training set) with L2-regularization, which allowed us to avoid overfitting and to deal with multi-collinearity among explanatory variables. The linear response kernel represents the response to instantaneous inputs ($\tau = 0$) of the continuous behavioral variable (Fig. 2c). We evaluated the prediction performance of the model using a correlation coefficient (Spearman's rank correlation coefficient: $r_s$) between the

spike density function of dCS train and the instantaneous spike probability of the model outputs in the remaining data (15%, a test set, Fig. 2b, d).

In the example cell in Fig. 2b, the prediction was correlated with the spike density function ($r_s = 0.48$, $p < 10^{-5}$, permutation test on dCS timing, one-sided). According to these kernels, an increase in spike probability preceded the lever-pull speed, whereas a decrease followed the lever-push speed. Therefore, dCSs of this cell were presumably motor- and sensory-related for lever pull and lever push, respectively. The mean prediction performance was $0.17 \pm 0.12$ (Fig. 2d) for all 517 cells in 9 imaging sessions. The model predicted dCSs in test sets with statistical significance ($p < 0.05$, permutation test on dCS timing, one-sided) in 352 cells (68%). This statistical criterion approximately corresponded to (mean $+ 1.65 \times$ standard deviations) of the prediction performance on the permutated dCS timings. The prediction performance in these cells was significantly higher than that of chance ($0.002 \pm 0.003$, from the permutation test on dCS timing, $p < 10^{-5}$, two-sided Wilcoxon signed-rank test). These results indicated that the model captured the response properties for the observed PCs. One possible reason for the moderate correlation was the low-rate and stochastic firing of dCSs. To further confirm the validity of our encoding model, we calculated the peri-event time histograms (PETHs) of the dCSs and the instantaneous spike probability of the model around the lever-pull events (Fig. 2e–h). PETHs derived from the dCSs in both training and test sets were very similar and highly correlated with the estimated values ($r = 0.94$, $p < 10^{-5}$; for example data in Fig. 2e–h), partly because the stochasticity of the spike responses was averaged out in the PETHs that reflected the spike probability. The mean correlation coefficient was $0.87 \pm 0.13$ (352 cells in the model with significant prediction performance, Supplementary Fig. 2b). Even for the test data, correlation coefficients between PETHs and their estimates were considerably higher than those between the single trains of dCS and the estimated instantaneous firing rates ($0.53 \pm 0.28$, compared with $0.17 \pm 0.12$, Supplementary Fig. 2c). Thus, the five variables could successfully predict the dCS around lever movement.

Task performance varied among the mice. Two mice performed the task better than others and obtained 44.5 (50 and 39 in two sessions) and 39 rewards during a session of 600 s (13.5 s and 15.3 s per reward). The encoding model for cell responses in these mice predicted significantly higher than by chance ($r_s = 0.13 \pm 0.07$, mean ± standard deviation; 143 cells, $p < 0.05$), suggesting that our analysis can similarly describe the responses of CSs during the task, regardless of the task performance of mice.

For further analysis, we used 352 PCs with dCS occurrences predicted to be statistically significant.

**Classification of the PCs according to the properties of CF responses.** We examined the linear response kernels of the model (e.g., Fig. 2c) to explore the property of the CF response to each behavioral variable. Thus, we applied cluster analysis to the response kernels to identify CF response properties that the cells shared within and among the imaging sites. The Gaussian mixture model clustering identified eight clusters (i.e., functional types, Fig. 3a), where the Bayesian information criterion scores acquired the minimum value (Fig. 3c). All were assigned to any one of the clusters. PCs in the individual functional types displayed similar response kernels with little variation. By contrast, the response kernels belonging to the distinct functional types differed from each other in the time courses and the combination of the behavioral variables to which the PCs were tuned (Fig. 3a, b and Supplementary Fig. 3). The kernels for pull speed, push

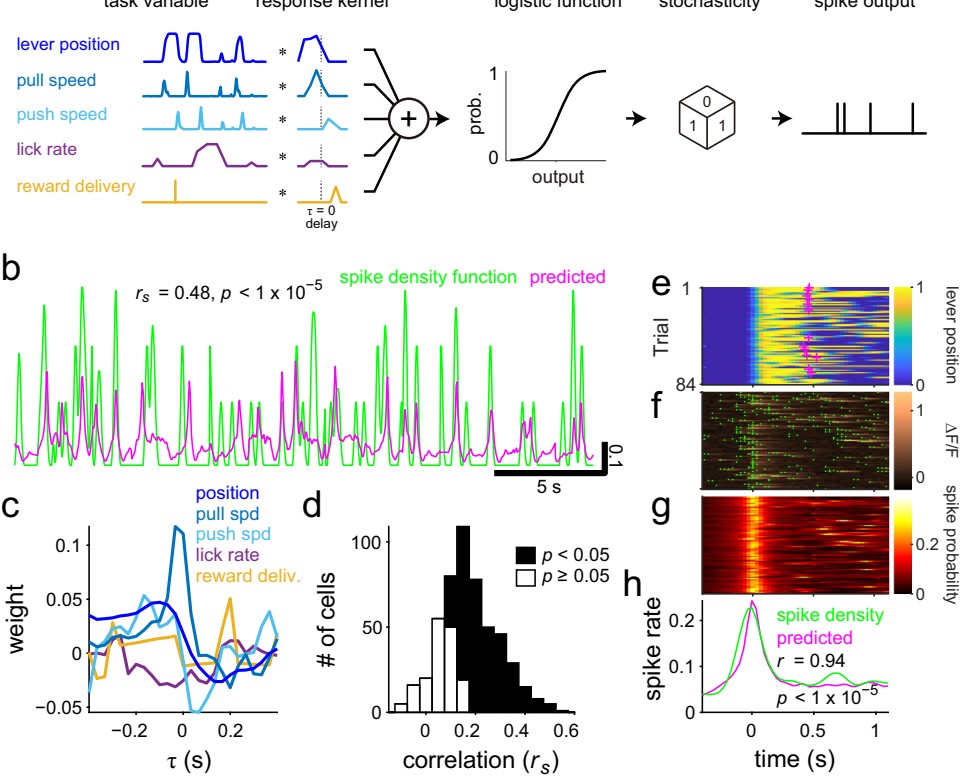

**Fig. 2 Characterization of climbing fiber (CF) responses to the behavioral variables by analyzing the encoding model. a** A schematic of the encoding model. **b** A prediction of the instantaneous spike probability (magenta) by the constructed model and the spike density function of a PC (green) corresponding to the model. Vertical scale bar: spike density and instantaneous probability. **c** Standardized linear response kernels of the constructed model in **b**. **d** Prediction performances of the models, Spearman's rank correlation coefficients between the spike density function and predicted spike probability in test sets ($n = 517$ cells). **e** Lever-pull aligned lever trajectory with task success timing (magenta cross). **f** Deconvolved spiking of the example cell (green dot) on the fluorescence change (ΔF/F) used for spike estimation. **g** Predicted instantaneous spike probability. **h** Event-mean of spike density and probability (PETHs).

speed, and reward delivery were similar to the counterparts of population-averaged PETHs (Supplementary Fig. 4). The modulations of the response kernels for lick rate seem to be dissimilar to those of PETHs for the lick bout start, which often coincided with other behavioral events (see Fig. 1e). Prediction performances of the encoding models were only marginally different across the functional types (Fig. 3d, $p < 10^{-5}$, one-way analysis of variance). Generally, the coefficients for lever movement (position, pull-speed, and push-speed) and lick rate were modulated for all functional types (99% bootstrapped confidence intervals deviated from 0, blue and purple lines in Fig. 3b, Supplementary Fig. 3). The modulation of PC activities to lever movements is consistent with the fact that the lobule V of the vermis/para-vermis is relevant for forelimb movements[43]. By contrast, the coefficient for reward delivery (yellow lines in Fig. 3b) revealed modulation in two functional types (99% bootstrapped confidence interval).

Types 1–4 and 7–8 PCs revealed an increased CF response with a negative delay for the lever position and the lever-pull speed. However, the timing differed among the types. CF responses in types 1, 2 and 7 PCs increased within 100 ms before the pull speed, whereas those in types 3, 4 and 8 PCs increased earlier (>300 ms before the pull speed). In type 4, the response became negative with a positive delay. In types 5 and 6 PCs, the responses displayed a sustained increase following the pull speed, thus indicating that the CF response probability in these types increased while pulling the lever. For the lever-push speed, CF responses in types 1, 2 and 5–7 PCs increased with different

timings around the push. Response in type 5 PCs increased before and after the lever push, whereas responses in type 1, 2, 7 and 6 PCs increased immediately before and after the lever push, respectively. In contrast, CF responses in types 1 and 4 PCs decreased around the push. The response in type 3 PCs also showed a sharp and negative peak, although the peak was narrower than the detection criterion. PCs of all functional types were modulated by the change in the lick rate (purple line, Fig. 3b). Although the modulations for the lick rate in type 4 PCs were similar to those for the lever movement in time (Pearson's correlation coefficient: 0.63–0.93, $p < 0.001$), those in other types were not. For reward delivery, CF responses in type 7 PCs transiently and prominently increased with a 200-ms delay, and those in type 3 decreased with a positive delay. In sum, CF responses in 23% (81/352) of PCs were found to change in response to both movements and reward delivery. Thus, the CF responses of PCs encoded two (lever movement and lick rate) or three (lever movement, lick rate, and reward delivery) different modalities of behavioral information. This was supported by CS responses to multiple modalities observed in population-averaged PETHs (Supplementary Fig. 4). In particular, PCs in type 7 responded to both lever pull and reward delivery.

**Spatial clustering of cells according to their CF response properties.** Spatial mapping of PCs in terms of the functional types revealed that the PCs classified as a certain functional type were distributed across imaging sites in different animals (e.g.,

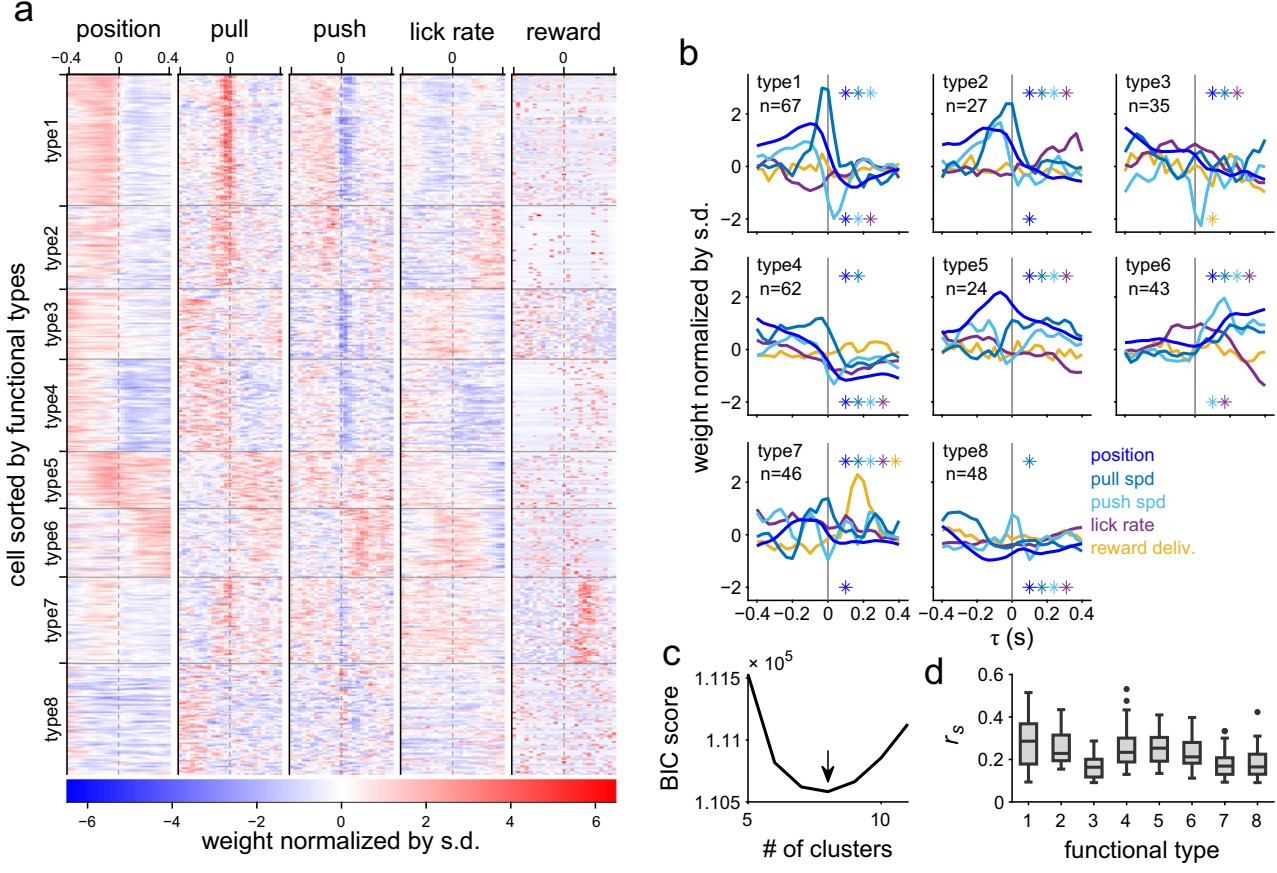

**Fig. 3 Functional types of cells according to the response properties to behavioral variables. a** Linear response kernels of the models for the individual cells sorted by their functional types (352 cells). The response kernels are normalized by their standard deviation. **b** Mean response kernels of the individual functional types. The asterisks in the first and fourth quadrants of each panel indicate that the mean of an indicated variable by their color significantly deviates from zero toward the positive or negative direction where the asterisks are located, respectively. **c** Bayesian information criterion (BIC) score against the number of clusters in the Gaussian mixture model displaying that the optimal number of clusters is 8, as indicated by the arrow. **d** Prediction performances of the cell across the functional types ($p < 10^{-5}$, $F(7, 344) = 14.5$, one-way analysis of variance). Box-plot elements: center line, median; box limits, upper and lower quartiles; whiskers, maximum and minimum values within 1.5× interquartile range; points, outliers.

type 1 cells in Fig. 4a, e, i and type 7 cells in Fig. 4c, d, h). Therefore, the CF response properties of the functional types were common for the lever-pull task at different imaging sites and in different animals. We observed the spatial clustering of PCs belonging to identical functional types (Fig. 4a–i). We found all types of clusters. Some types of clusters were in different imaging sites. The frequency that two PCs located closely (<50 μm) in the mediolateral direction belonged to the identical functional type was higher than that by chance in eight of nine imaging fields ($p < 0.05$, permutation test, two-sided, Fig. 4j). Moreover, we observed sharp borders between the functional clusters (Fig. 4c–e, i), and the overlap between clusters was within the width of few PCs in the majority of cases. We measured the lateral width of the two clusters, which appeared to be included in the imaging sites entirely. The lateral width of the type 3 cluster and type 6 cluster in Fig. 4c, e as 163 μm and 349 μm, respectively. Thus, the PCs were spatially clustered according to their CF response properties to the behavioral variables, which were a combination of the variable types (lever, lick, and reward) and their temporal patterns.

As expected from the spatial clustering of PCs belonging to the identical functional class, adjacent PCs along the mediolateral axis often share similar response kernels underlying the functional classification. For example, cells #1–3 in Fig. 5a displayed clear positive responses to the lever-push speed around

a time lag of 0 (Fig. 5a). By contrast, cells #4 and #5 responded negatively to the lever-push speed. Cells #6–8 showed prominent responses to the reward delivery with a short delay. The larger the distance between the two PCs, the lower the similarity between their response kernels (Fig. 5b–d). The degree of similarity (Pearson's correlation coefficient) between the response kernels of the two cells was negatively correlated with the mediolateral distance between them ($r = -0.76$, $p < 10^{-5}$, $n = 2278$ pairs, Fig. 5e). We obtained similar findings for seven out of remaining eight imaging sites ($r$ range, $-0.35$ to $-0.70$, $p < 0.01$ in seven sites, $r = -0.01$, $p = 0.8$, in one site) and the overall population ($r = -0.41$, $p < 10^{-5}$, $n = 8544$ pairs, Fig. 5f). In other words, the PCs were spatially arranged along the mediolateral axis independent of the functional classification, and neighboring PCs displayed high signal correlation, i.e., similar CF response profiles to behavioral variables. The regression line for the signal correlation and mediolateral (ML) distance between cells with <200 μm of ML distance (Fig. 5f) indicated that the response properties of two cells with approximately 200 μm of ML distance were dissimilar to each other. In addition to the dependence on the mediolateral distance, we observed a sharp increase or decrease in the signal correlation between PCs along the mediolateral axis (Fig. 5b, c). Thus, functionally similar PCs with identical CF response properties formed spatially segregated functional clusters.

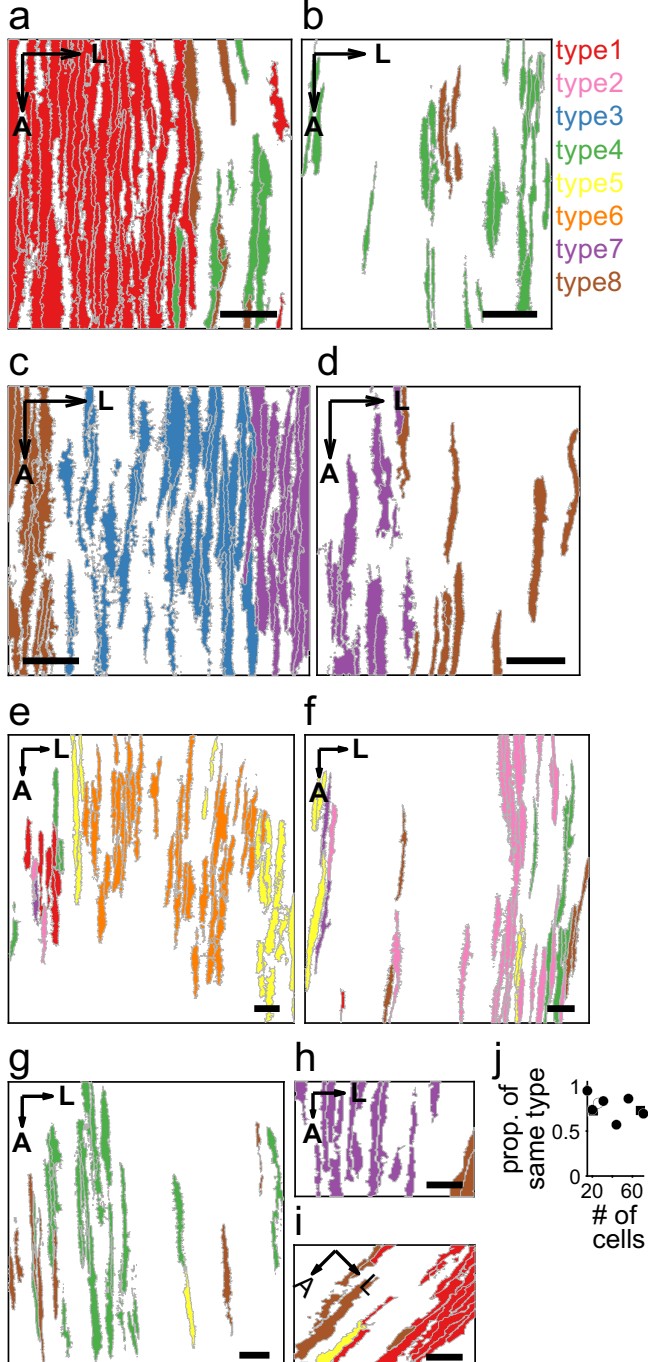

**Fig. 4 Spatial clustering of PCs according to the functional types of response properties to the behavioral variables. a–i** The distribution of cells according to the functional types across the cortex. Each color denotes the identity of the functional type. Panels **c** and **d** are the different FOVs from the same mouse, and all other panels are from different mice. A: anterior. L: Lateral. Scale bars: 50 μm. **j** The proportion that two cells within 50 μm of the mediolateral distance belong to the identical functional type was plotted against the number of cells in the nine sites. Closed dots indicate the imaging sites whose proportions are significantly higher than the chance level (*p* < 0.05, permutation test, two-sided, 5000 permutations). Square dots indicate different imaging sites in the same animal.

**Spatial clustering of cells sharing noise fluctuation in CSs.** Lastly, we examined the relationship between the spatial clusters of PCs and their functional connectivity by analyzing noise correlations, calculated as the correlations in the event-to-event

fluctuation of dCSs between two PCs. CF inputs to PCs induce CSs reliably; therefore, noise correlations in dCSs reflect the degree of functional connections between neurons in the inferior olivary nucleus. We analyzed the dCSs around the lever-pull onset and subtracted the model predictions from the dCS. An example cell in Fig. 6a (black) showed a high noise correlation with adjacent cells, and the correlation decreased with the mediolateral separation between cells. The spatial pattern of the noise correlation magnitude in the imaging field was similar to that of the signal correlation of the PCs to the identical cell (Fig. 5b). The matrix of noise correlations (Fig. 6b) also demonstrated a similar pattern to that of signal correlations in Fig. 5d ($r = 0.54$, $p < 10^{-5}$). In five imaging sites, noise correlations were positively and significantly correlated with signal correlations ($p < 0.05$). Moreover, we observed a sharp boundary in the magnitude of noise correlation (Fig. 6a, b) at a position similar to the boundary of the signal correlation (Fig. 5b, d), thereby suggesting a higher noise correlation between adjacent cells belonging to the identical functional type. To test this notion, we compared the noise correlation between pairs of PCs within and across functional classes. We computed the noise correlations of PC pairs separated mediolaterally by <100 μm to control for the effect of distance. Noise correlations of pairs belonging to identical functional clusters were higher than those belonging to different functional clusters (Fig. 6c, $p < 10^{-5}$, two-sided Wilcoxon's rank-sum test). In five imaged fields, we observed higher correlations for PC pairs belonging to the identical functional clusters (Fig. 6d, $p < 0.05$, two-sided Wilcoxon's rank-sum test). Taken together, PCs encoding similar information shared noise fluctuations in CF responses, which were presumably based on gap junctional coupling between the inferior olivary neurons that encoded common behavioral variables (Fig. 6e).

## Discussion
In the present study, we reported on multiplexed information conveyed by individual CF inputs to the cerebellar PCs during voluntary movement in mice. First, using two-photon calcium imaging, we showed that a self-initiated forelimb lever-pull task-induced dendritic calcium signals of PCs, which represented CSs, in the forelimb region of the lobule V of the cerebellar vermis. Second, the CS activity of each PC was reliably predicted by an encoding model constructed using five behavioral variables (lever position, lever pull and push speed, lick rate, and reward delivery). The CS activity of PCs encoded lever movements; nonetheless, it also encoded the instantaneous lick rate and reward delivery. Third, the cluster analysis revealed that the PCs were classified into eight groups with different response properties. Fourth, neighboring PCs shared CF response properties; PCs displaying similar response patterns were spatially clustered and delineated by sharp boundaries. The CS activity of PCs in the same cluster covaried more strongly than that in different clusters. These results indicate that individual CF inputs to the cerebellum convey multiplexed information about different aspects of voluntary movement, including reward-related information. Locally clustered CF inputs carrying similar information form a foundation of the microzonal organization of the cerebellar cortex.

The self-initiated lever-pull task[39–42] offers several advantages for investigating multiple behavioral variables. It allows quantitative measurements of lever movements, the detection of licking behavior, and reward delivery. Moreover, it is suitable for exploring the relationship between neural activity and motor behavior. This is because the trajectory of the lever in this task displayed considerable variability in kinematic parameters (amplitude, velocity, holding duration, and interval) (Fig. 1). However, a recent study demonstrated that task-unrelated

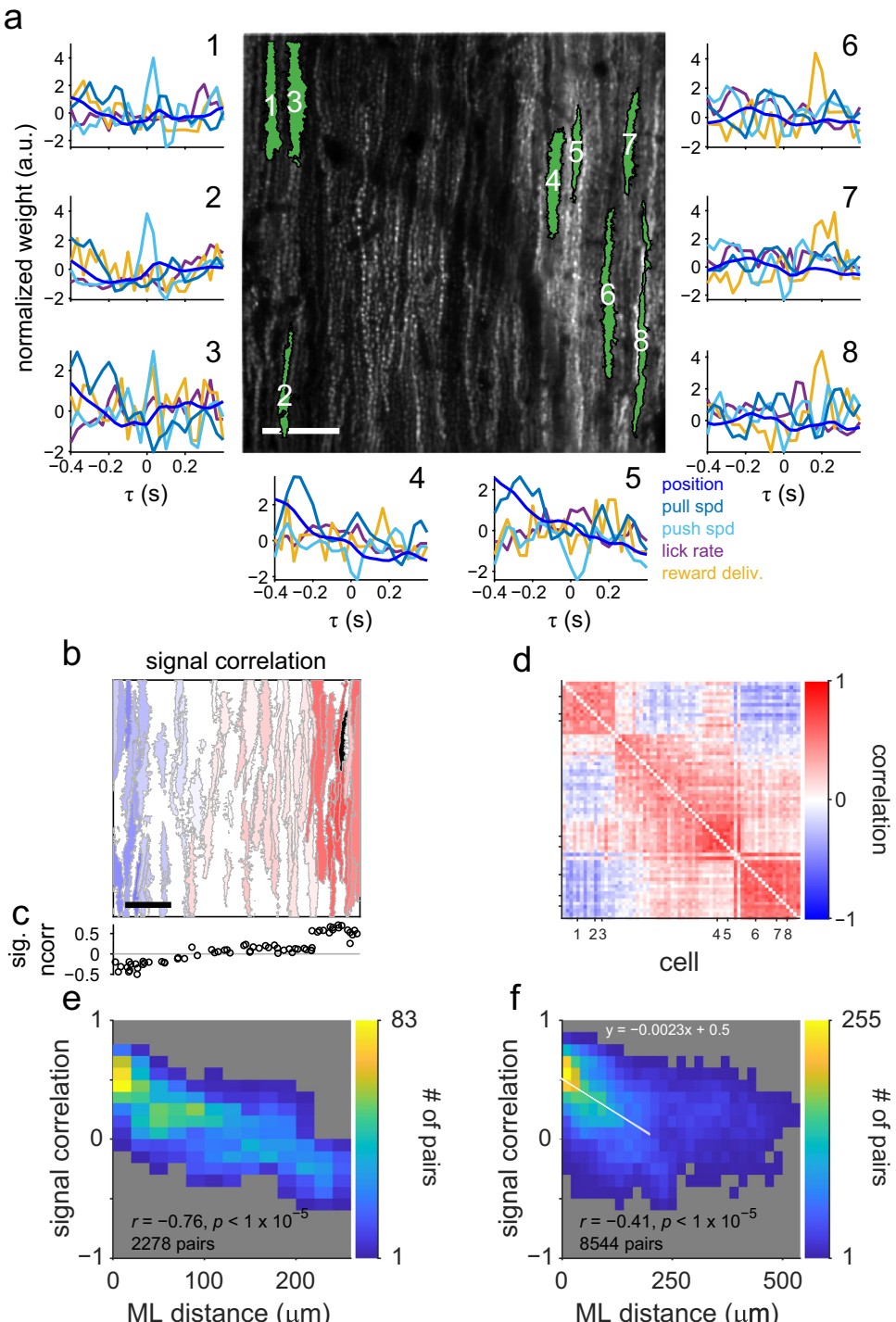

**Fig. 5 Spatial clustering of cells with similar response kernels. a** Linear response kernels in an example site. The numbers on the plots correspond to those in the center image. Scale bar: 50 μm. **b** An example of a spatial map of signal correlations (correlation coefficient between the response kernels of two cells). The color indicates a signal correlation of a cell with the black-colored cell (#7 in **a**). The color scale is the same as shown in **d**. Scale bar: 50 μm. **c** Signal correlations between the black-colored cell and other cells against the mediolateral position, aligned with the spatial map. **d** A signal correlation matrix of the imaging plane in **a**. Each row or column represents signal correlations between a particular cell and other cells. The cells are sorted by a mediolateral position in the imaging region. The numbers on ticks indicate the cell number in **a**. **e**, **f** The relationship between the mediolateral (ML) distance and signal correlation. **b**, **c**, **d**, **e** Data from an example imaging site displayed in **a**. **f** Population data. The white line and shade in the panel are a regression line for the pairs with <200 μm of ML distance ($n = 7188$) and its standard error.

movements substantially affect neural dynamics during a task[53]. This necessitates further investigation to completely explain which behavioral parameters are encoded in CSs by monitoring whole-body movements (e.g., whisking, other limbs, trunk, and tail movements)[54] besides forelimb lever movements.

The encoding model analysis elucidated the relationship between sensory stimuli and/or behavioral variables and neural activities in various brain regions[53,55,56]. The brain receives a substantial amount of diverse information from the external world and represents the information by the activity of billions of

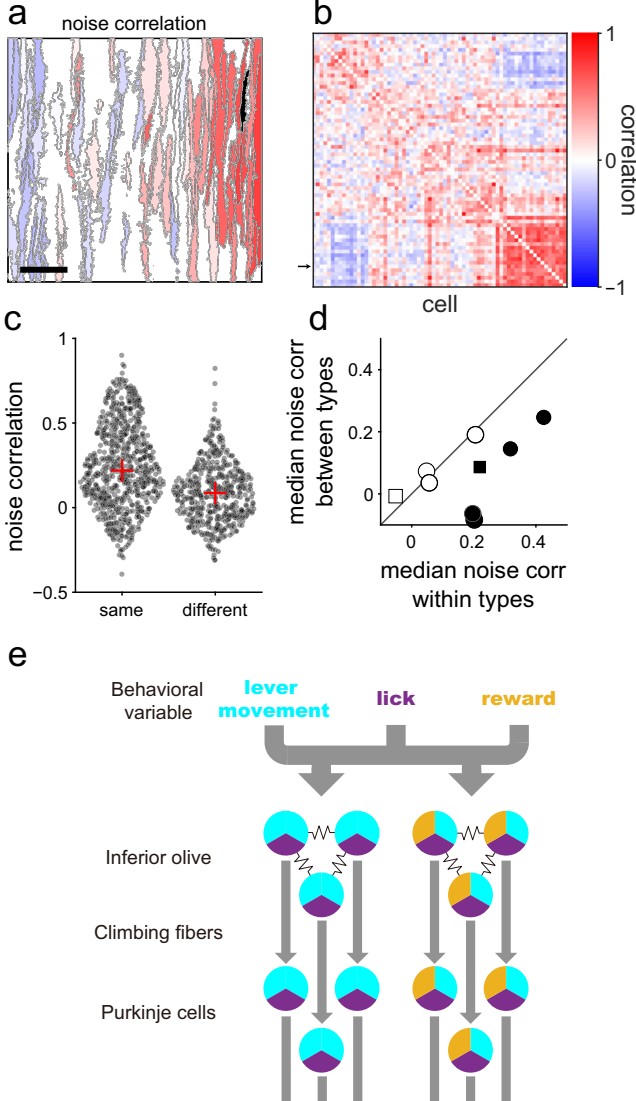

**Fig. 6 Noise correlation between complex spike (CS) responses of cells.**
**a** An example map of noise correlation of cells with a representative cell (black). The cell is identical to that displayed in Fig. 5b. Scale bar: 50 μm. **b** A noise correlation matrix of an imaging region displayed in **a**. The cells are put in the same order as in Fig. 5d. The arrow indicates the black-colored cell in **a**. **c** Noise correlations for cell pairs in the example map displayed in **a**. Left: cells are from identical functional types ($n = 739$ pairs). Right: Cells are from different functional types ($n = 516$ pairs). Red cross: median. $p < 10^{-5}$, Wilcoxon's rank-sum test. **d** Median noise correlations of cell pairs from identical or different functional types in 9 imaging regions. Closed dots represent a statistically significant difference between the data from identical and different types ($p < 0.05$, two-sided Wilcoxon's rank-sum test). Square dots indicate that the imaging sites were different sites in the same animal. **e** A schematic illustration summarizing our findings.

neurons. Thus, the parameters of behavior represented by individual neuronal activities are unclear, and it is usually difficult to determine a relationship between neural activity and behavior. The encoding model analysis is a computational method that aims to construct a quantitative model that explicitly transforms a combination of multiple complex external variables into observed neural activities[51,55]. Although the CS activity of PCs is at a relatively low rate (~1 Hz on average), the encoding model analysis has been successfully applied to determine the relationship

between the CS firing pattern and animal behavior, such as eye movements[8] and limb movements[9]. In this study, the encoding models for 68% of the PCs examined significantly predicted the CS activities.

The prediction performance on dCS occurrence in single trains was moderate ($r_s = 0.17 \pm 0.12$, mean ± standard deviation, Fig. 2d). The moderate correlations may have been caused by at least two reasons. First, our models contain only five kernels, although they had 25-time points individually. These kernels were probably insufficient to explain the CS activities completely. Other unmonitored or unknown parameters, including motor, sensory, and task parameters, are expected to contribute to the CS activities. However, a complete explanation of the CS activities was not absolutely necessary for our conclusion, namely the joint encoding of multiple behavioral parameters by CSs. Second, the lower prediction performance of the encoding model was partly caused by the low firing of CSs (<2 Hz). Our encoding model predicted the dCS firing (0 or 1) as the probability ranging from 0 to 1. Even if the firing rate of dCSs to a given behavioral parameter truly was a continuous variable from 0 to 1, the dCS firing takes 0 or 1 at a particular time. This firing property should lower the correlation between dCSs and their prediction. To predict the dCS firing probability correctly, we need repeated trials. In Fig. 2e–h, we showed that the encoding model predicted the dCS probability for the lever pulls well. This result indicates that the models describe the activities around the lever-pull behavior well.

CF inputs to individual PCs in the lobule V of the cerebellar vermis encode lever movements. This is consistent with previous studies showing that this region is related to forelimb movements in rodents[37,43] and that CS activity of PCs represents limb kinematics[7,9,10]. The lever-related CS signals may represent both motor signals and sensory feedback signals. Streng et al.[10] mentioned that CS discharge in monkeys performing a pseudo-random cursor tracking task predominantly represents predictive motor signals, not sensory feedback. Similarly, our encoding model and clustering analysis revealed that CS activity related to the lever-pull speed (Fig. 3a, position and pull) preceded the lever pull (types 1, 2, and 7). Therefore, these CS activities represented motor signals regarding pulling the lever, which involves movements of the forelimb as well as other body parts, including the face, hindlimb, and tail[54]. In addition, we observed earlier CS activity (>300 ms before lever pull) in three types (types 3, 4, and 8, Fig. 3a). These signals were temporally broad and not precisely time-locked to lever-pull movements. Thus, they may not be directly related to the lever pull but were presumably activated by other factors, such as the animal's attention to prepare or initiate movements. The lever-push speed also modulated CS activity in most PCs (Fig. 3a, push, decreased in type 1, 3, 4, and 6 cells; increased in type 1, 2, 5–7 cells). In our task device, the lever was loaded by a magnetic force; thus, the lever-push movements were likely to be automatic return movements of the pulled-lever, not the active pushing-back movements of the animal. Therefore, most of these CS activity modulations may be related to sensory feedback, not motor signals. Indeed, responses related to lever push peaked at a positive delay in the push (Fig. 3a, b).

Although the majority of PCs in the lobule V of the cerebellar vermis primarily encoded sensorimotor signals regarding lever movements, a fraction of the PCs (type 7) were strongly activated immediately following reward delivery (Fig. 3a). In successful lever movements, the mice obtained a water reward following a holding period of 400 ms and immediately began an intense licking bout (Fig. 1e). However, PCs modulated by the reward delivery display only a small modulation for lick rate (Fig. 3a, b, lick rate). Therefore, these signals were likely reward-related. PETH of these cells also shows a clear peak 0.2 s after the reward delivery (Supplementary Fig. 4). As the signals emerged after the

reward delivery, they were unlikely to be reward anticipation signals. Previous studies have reported the presence of reward-related CF signals in the cerebellar hemisphere (lobule simplex, crus I, and II)[13–17,57]. Our findings revealed that reward-related CF signals also exist in the vermis, where reward-related activities were observed in granule cells[58]. To our knowledge, there have been no reports of direct projections from reward-related regions to the inferior olivary subnuclei that send CFs to the cerebellar vermis. Reward-related CF responses may originate from yet unknown projections conveying signals from reward-related regions or feedback circuits involving the cerebellar nuclei, inferior olivary nuclei, and associated PCs that receive reward-related signals from the mossy fiber–granule cell–parallel fiber pathway. Considering that the CS discharge during eyeblink conditioning is similar to the activity of midbrain dopamine neurons during reinforcement learning[59,60], CF signals can encode temporal-difference prediction errors[18]. Similar mechanisms can be applied to reward-based learning in the cerebellum[61,62]. However, a recent study demonstrated that PCs during reinforcement learning encode error signals in simple spikes but not in CSs[63]. Therefore, CF inputs in reward-based learning may not be an error signal that drives learning. Our findings suggest that CSs are activated during rewarded movements. Thus, the CF inputs may represent positive reward outcomes, not error signals.

PCs exhibiting reward-related signals also displayed lever movement-related activity (Fig. 3, Supplementary Fig. 3 and Supplementary Fig. 4), indicating that they encoded multiple behavioral variables related to both motor and non-motor functions in their CS activities. Multiplexed movement-related information encoded in CS activity has been observed in the oculomotor cerebellum of monkeys during ocular following responses[8] and performing saccadic eye movement task[64], showing that errors and kinematics related to eye movements and task parameters are encoded in CS activity. Furthermore, we show similar multiplexed encoding in CS activities of PCs in the cerebellar hemisphere (Crus II) during Go/No-go learning[65]. Given that reward-related CF signals are distributed widely over the cerebellar cortex[13], multiplexing movement- and reward-related information in a single channel is one of the general properties of the CF signal in many cerebellar regions. Multiplexed movement- and reward-related information in a single CF may be interpreted by the PC in combination with simultaneously incoming parallel fiber inputs and inhibitory inputs from molecular layer interneurons. This may allow both action-based and reward-based behavioral learning in a single PC.

If the CS activities encode reward prediction error, not movement-related information, CS activities may increase in the period of lever movement and decrease or may not change after the reward presentation in well-trained mice. However, the firing rate of dCS in our results increased after the reward delivery in two mice with higher performance. Moreover, in other lower-performance mice, strong responses were observed around lever movement. Therefore, it is unlikely that the activities signaled the reward prediction. On the other hand, cells in type 3 increased activities before lever-pull strongly and reward delivery timing weakly. We cannot rule out the possibility that these CS activities during the lever movement have signals regarding reward prediction error. To examine whether CS activities during the lever-pulling period encode reward prediction errors, we need to record the activities successively throughout the learning process or manipulate the probability of reward occurrence.

Simple spikes (SSs) of PCs cannot be detected by $Ca^{2+}$ imaging technique because they rarely generate dendritic $Ca^{2+}$ signals[66], but they are of course in the output from PCs. CS and SS firing rates are often reciprocal: when the CS rate increases, the SS rate decreases[67], although the reciprocity depends on species, experimental paradigms and cerebellar regions[68]. Furthermore, granule cell activity encodes reward-related signals in the same cerebellar region where we observed[58]. Therefore, it is possible that SS also encodes behavioral variables and multiplexes movement- and reward-related signals.

Functionally similar CF inputs were located close to each other and were spatially clustered along the mediolateral direction (Figs. 4 and 5). Each cluster encoded lever movements, licking behavior, and reward delivery with different combinations and temporal patterns (Figs. 3 and 4). The width of the clusters was on the order of 100 μm, consistent with that of microzones[22,69] (Fig. 4). Therefore, CF inputs to each microzone encoded the parameters of animal behavior differently. PC axons extensively converge onto the target neurons in the cerebellar nuclei[70], and the synchronous firing of PCs (Fig. 6) could elicit spiking in nuclear neurons; therefore, CSs from different microzones can converge and be transformed into the spiking of nuclear neurons, which encodes behavioral parameters, such as lever trajectory or corresponding motor output. We intend to conduct further experiments to determine if the encoding of behavior in each CF afferent cluster is innately determined or acquired through learning.

The number of functional types reported in this study, eight, is not definitive. Because the individual imaging sites cover only a part of lobule V left vermis, we observed several functional clusters in individual sites. We could not examine how many functional types were represented in individual mice. It is possible that the difference in behaviors among mice was reflected in the functional types observed in individual mice. It is required to record the activities of cells in a sufficiently large area of lobule V and examine their response properties to a wide repertoire of behavior of mice to reveal the functional architecture in lobule V more comprehensively.

## Methods

**Mice**. All experiments were approved by the Animal Experiment Committees of The University of Tokyo and University of Yamanashi. We used adult (>3 weeks) male C57BL/6J wild-type (WT) (C57BL/6JJmsSlc, Japan SLC, RRID:MGI:5488963) or PCP2 (L7)-Cre mice (Tg(Pcp2-cre)2Mpin/J, The Jackson Laboratory, RRID:MGI:3531210). The mice were kept in a reverse-phase 12-h/12-h light-dark cycle. Food was provided *ad libitum* throughout the study. Water intake was restricted from 2 days before commencing the behavioral training period. Following the regular task training, they were provided with 1% agarose gel so that the total daily water intake was 1 g to maintain their body weight.

**Surgery and virus injection**. The mice underwent surgery for the cranial window and virus injection in a day[17]. We intraperitoneally administered 15% D-mannitol (4.5 mg/g) dissolved in phosphate-buffered saline to increase the efficiency of the viral vector infection[71]. The mice were anesthetized with isoflurane (4–5% for induction and 1–2.5% for maintenance). We applied a local anesthetic (Xylocaine jelly 2%, Aspen) to the skin over the skull. Subsequently, we exposed and coated the skull with dental adhesive acrylic resin (Superbond, Sun Medical). A customized metal head plate with a hole (diameter: 3.8 mm) was fixed with a light-curing resin (Panabia F 2.0, Kuraray) for head fixation and imaging. We created a 3-mm cranial window (center at 0.5 mm lateral and 6.5 mm caudal from the midline and bregma, respectively). These windows covered the forelimb area of the lobule V on the left vermis. We injected an adeno-associated virus (AAV) encoding a calcium indicator protein, GCaMP6f

(Chen et al., 2013) (AAV2/1.CAG.Flex.GCaMP6f.WPRE.SV40 in L7-Cre mice or a mixture of AAV2/1.CAG.-Flex.GCaMP6f.WPRE.SV40 and AAV2/1.CMV.PI.Cre.rBG for WT mice, obtained from Penn Vector Core or Addgene) into the lateral part of the left vermis lobule V. We injected 300–500 nL of AAV at a rate of 20 nL/min at a depth of 200–250 μm from the pial surface. The glass pipette was kept in place for 5 min post-injection. A round coverslip (Matsunami glass) was placed over the craniotomy site, and the edge of the coverslip was sealed with a tissue adhesive bond (ethyl cyanoacrylate, Aron Alpha A Sankyo, Toagosei) and dental acrylic resin (ADFA, Shofu). We applied an antibiotic ointment (gentamicin sulfate 0.1%, Iwaki Seiyaku) over the incision to reduce infection. Following surgery, the mice were individually housed in a cage. We did not observe any abnormal behavior following the AAV injection.

The injection of a mixture of AAVs encoding GCaMP6f and Cre in WT mice resulted in GCaMP6f expression in PCs and some molecular layer interneurons. Considering variations in the shape of the cell body and fluorescence response kinetics of molecular layer interneurons from those of PC dendrites[72], we discriminated between the two types of cells and excluded the molecular layer interneurons from the analysis (refer to Imaging data processing).

**Lever-pull task**. We commenced the task training at least 10 days following the surgery and virus injection to ensure complete recovery and GCaMP6f expression. Water restriction was initiated two days before beginning the habituation and task training. On day 1, we set the mice to the task apparatus (O'Hara) without a lever-pull task. From day 2, they were subjected to a lever-pull task. One tip of a 180-mm-long lever was positioned 10 mm below the left side of their mouth. The lever could revolve in the horizontal plane around the point 120 mm apart from the tip. The mice grasped the lever around the tip and pulled it caudally with their left forepaw (Fig. 1a). The maximum lever displacement from the resting position was 5.5 mm. We symmetrically placed a fixed bar on the right side, thus allowing the mice to place their right forepaw on it.

We loaded a magnetic force on the lever to return it to the maximum forward position (i.e., the resting position). Therefore, the mice were required to overcome the magnetic force to pull it. Upon loosening the pulling force, the lever automatically returned to the resting position (passive push). We did not determine whether a forward movement of the lever was caused by an active or passive push. We monitored its position using a magnetic sensor (HA-120, MACOME). We set the lower and higher thresholds of the lever displacement at 10% (0.55 mm) and 83% (4.57 mm) of the maximum from the resting, respectively. Mice had to retain the lever position below the lower threshold for >1 s and subsequently pull and maintain it beyond the higher threshold for a certain period (hold-duration; range: 0–400 ms) to obtain 4–8 μL of the water reward (success). The water reward was delivered to mice from the water port in front of their mouth, 380 ms following task completion. They could lick the water port freely at any time during the task. Licking was detected using an infrared photo beam sensor (O'Hara).

The mice were trained for 1 h per day. At the beginning of the lever-pull training, they obtained a reward only by pulling the lever beyond the higher threshold (hold-duration = 0). We gradually increased the hold duration to obtain a reward with an increase in the success frequency. Upon achieving a hold duration of 400 ms with a success frequency of 2 times/min, we performed two-photon calcium imaging on the following day. We recorded behavioral data, namely lever position, licking timing,

and task success timing (380 ms before reward delivery timing), at 200 Hz. We calculated the instantaneous licking rate at each licking timing $T_n$ (timing of nth licking) from a series of three licks as follows[72],

$$\text{LickingRate}(T_n) = \frac{3}{T_{n+1} - T_{n-1}} \qquad (1)$$

We linearly interpolated these data except reward delivery timings and downsampled them at the frame acquisition time of two-photon imaging. Success timings were assigned to frame acquisition times closest to them. The instantaneous licking rate below 2 Hz was set to zero. Task control and data acquisition were performed using a custom-written program in the LabView software (National Instruments).

**Two-photon imaging**. We performed two-photon calcium imaging from the dendrites of the cerebellar PCs expressing GCaMP6f in task-performing mice. We used a two-photon microscope (MOM, Sutter Instruments) equipped with a 25× objective lens (Olympus) and Ti:sapphire laser (Mai Tai HP DS, Spectra-Physics). The microscope was controlled using Scan-Image software[73] (Vidrio Technologies). We used an excitation wavelength of 900 nm. An 8 kHz resonant scanner and a galvanometer scanner were used for scanning in the x- and y-axis directions, respectively. The imaging fields were as follows: $275 \times 275$ μm² (four fields, three mice), $550 \times 550$ μm² (two fields, two mice), $640 \times 640$ μm² (one field, one mouse) with a resolution of $512 \times 512$ pixels, or $180 \times 270$ μm² (two fields, two mice) with a resolution of $128 \times 128$ pixels. Images of $128 \times 128$ pixels were acquired at 109 Hz for 550 s (60,000 images), whereas all other images were acquired at 30 Hz for 600 s (18,000 images).

**Imaging data processing**. We extracted the regions of interest (ROIs) using the Python version of Suite2P software[45]. This program performed a two-dimensional phase-correlation-based non-rigid image registration in the XY plane (block size: $128 \times 128$ pixels), extracted the ROIs considered as PC dendrites, and calculated the fluorescence intensity changes in each ROI. The parameters for ROI extraction were 40 and 10 for the "diameter" (diameter of PC dendrites) and 0.3 for "tau" (the decay time constant of calcium indicator). We regarded the individual ROIs that met the following criteria as the dendrites of individual PCs: (1) considerably large ROI (>100 μm²) as PC dendrite and longitudinally shaped in the rostrocaudal direction (>1.65, a ratio of two standard deviations of the 2D-Gaussian function fitted to the ROI) and (2) displaying a transient increase in the fluorescence intensity five times larger than the standard deviation of the fluorescence. Two ROIs with a mediolateral distance <5 μm and a correlation coefficient of their fluorescence traces >0.75 were considered dendrites from the identical cell and combined into one ROI. Using the aforementioned correction, 825 (56%) of the 1,478 initially extracted ROIs were subjected to further analysis (Fig. 1d).

Previous calcium imaging studies have demonstrated that CF inputs to a PC trigger widespread calcium transients in the entire dendrite[44]. Therefore, the responses between ROIs derived from the identical cell are likely to be highly correlated. However, we could not exclude the possibility that the two rostrocaudally adjacent ROIs displaying low correlation (<0.75) originated from a single cell because of the high recording noise in individual ROIs.

We observed prolonged fluorescence changes that lasted for several seconds in some ROIs. Generally, the firing rate of CSs in PCs is approximately 1 Hz[47,49], and CS-induced dendritic

calcium transients monitored using GCaMP6f display a rapid rise and decay time constant of several 100 ms[33]. Therefore, these prolonged fluorescence changes are unlikely to reflect the actual CSs. We discarded data during the period when the baseline fluorescence increased by >20%. The baseline fluorescence was estimated by smoothing the traces by performing a robust locally weighted linear regression (window size = 2.5 s). In parallel, the fluorescence waveform was high-pass filtered (0.5 Hz) to remove slow fluctuations and trends in the baseline caused by focus drift and photobleaching. We performed deconvolution of the fluorescence and obtained binary spike trains using the OASIS program[46]. Parameters in the OASIS were manually selected to assign spikes to fluorescence transients and not noise fluctuations (Fig. 1e).

**Encoding model analysis**. We performed an encoding model analysis to quantitatively examine the response properties of CSs in PCs in terms of behavioral parameters. We used a generalized linear model with Bernoulli distribution to regress the binary deconvolved CS data (Figs. 1e and 2a). The model inputs were the lever position and speed, the instantaneous licking rate, and the water reward delivery. The occurrence of CS was the model output. The lever speed was separately converted to backward and forward speeds (pull speed and push speed, respectively) because CSs may discretely encode them. We included the water reward delivery as an input variable, despite the lowered prediction performance of the encoding model following the inclusion of reward delivery (Supplementary Fig. 2a). This is because its contribution may have been underestimated owing to fewer occurrences of reward delivery events than those of other variables (Fig. 1b). The task failure timing was the timing 780 ms after the period of lever holding started in the failed task (Supplementary Fig. 1). Inclusion of the task failure timing did not increase the prediction performance, but rather slightly decreased it ($p = 0.0001$, Wilcoxon's signed-rank test, Supplementary Fig. 2a). Therefore, the results suggested that the dCS were not modulated by the reward-related error, or not modulated strongly sufficient to be detected. The model used was as follows:

$$y(t) = \sum_{i}^{5} \int_{t-400}^{t+400} x_i(\tau) r_i(t - \tau) d\tau + b_0 \quad (2)$$

$$p(t) = \frac{1}{1 + e^{-y(t)}} \quad (3)$$

where $x_i(t)$ was each behavioral input at time $t$. Inputs were lever position, pull speed, push speed, lick rate, and reward timing. $r_i$ was the response kernel for each model input. $b_0$ was an intercept. $p(t)$ was spike probability ($0 \le p(t) \le 1$). We downsampled data acquired at 109 Hz from two imaging fields to 30 Hz. We extracted and concatenated the data from 0.4 s before the onset of individual lever pulls to 0.7 s following its return to the resting position. The five behavioral variables were individually z-scored and temporally filtered by the response kernels, linearly summed with an intercept, and converted to the probability of CS occurrence using a standard logistic function. The response kernels spanned from −400 ms to 400 ms to capture motor-, sensory-, and reward-related signals.

We fit the model output to the deconvolved CSs (dCS) by optimizing the intercept and five response kernels, which represented the response properties of CSs to the individual behavioral variables. For model construction and accuracy evaluation, the behavioral variables and dCS data were divided into 100 chunks. We randomly selected 85 chunks as the training set, whereas the remaining 15 chunks were used for model evaluation (test set). Fitting was performed in the training sets of

individual neurons using the maximum likelihood estimation with L2-regularization, which allowed us to avoid overfitting by simplifying the response kernels. Moreover, L2-regularization deals with multicollinearity among explanatory variables, although the behavioral variables were weakly correlated ($0.09 \pm 0.09$, mean ± standard deviation for absolute values of correlation coefficients. The mean values ranged from 0.07 to 0.10 across mice). The loss function, which was minimized to fit the model, was as follows,

$$\begin{aligned}
\text{loss function} = -\sum_t [dCS(t) \log p(t) \\
+ \{1 - dCS(t)\} \log\{1 - p(t)\} + \lambda \sum_i \|\boldsymbol{w_i}\|^2
\end{aligned} \quad (4)$$

where $\boldsymbol{w_i}$ was a set of coefficients of the response kernel of each input ($i$), $\|\cdot\|$ denotes the Euclidean norm. For individual neurons, we determined the regularization parameters, $\lambda$, using 10-fold cross-validation. Regularization parameters for the minimum average deviance were used for the models. The final models were obtained using all data from a training set and regularization parameters. The prediction performance of the final model was evaluated using Spearman's rank correlation coefficient between the spike density function obtained from the dCS train (standard deviation of Gaussian kernel = 66 ms) and the predicted spike probability in the test set. For the analysis, we constructed models for cells >50 dCSs during the extraction period. Moreover, we removed the cells with a relatively short extraction period of data in individual imaging regions after removing the periods where cells showed prolonged fluorescence change. Finally, we obtained 517 cells. Models with a prediction performance higher than chance levels estimated from models using randomized dCS timings ($p < 0.05$, permutation test, one-sided, 2,000 permutations) were used for analyzing the response properties (352/517).

**Peri-event time histograms (PETHs)**. To confirm the validity of our encoding model, we calculated the PETHs of the dCSs and the instantaneous spike probability of the model around the lever-pull events with resting periods >400 ms (Fig. 2e–h). To evaluate the similarity, we calculated Pearson's correlation coefficients between the event means of actual spike density and instantaneous spike probability.

**Classification of cells based on response properties**. We classified the cells based on the shapes of their response kernels by cluster analysis with a Gaussian mixture model (GMM) fitting[74]. We performed clustering on the response kernels using the MATLAB function "fitgmdist". Covariance matrices for individual Gaussians in the GMM were diagonal and unshared among them. The estimated parameters of clustering for a particular number of clusters may depend on the initial GMM parameters; therefore, we repeatedly performed clustering by randomly changing the initial parameters 6,000 times. We adopted the clustering that revealed the maximum likelihood for the specified cluster number. We determined the number of clusters with the lowest Bayesian information criterion (BIC) score. Each cell with a posterior probability >0.7 was assigned to its corresponding cluster because the probability must be the largest among the clusters.

Statistical tests were performed for the linear response kernels to individual behavioral parameters in individual functional types. We calculated 99% bootstrapped confidence intervals of the mean. When the 99% confidence interval of the mean was >0 (or <0) for 150 ms continuously, the mean of the kernel was regarded as significantly modulated.

**Population PETH**. We computed population-averaged PETHs for individual functional types and behavioral events. The events were "pull", "push", "lick bout", and "reward delivery". "Pull" was a lever movement where the lever moved from the resting position (time = 0) beyond the higher threshold. "Push" was a lever movement where the lever moved from the holding position (above the higher threshold, time = 0) to the resting position. "Lick bout" was the sequential licking (inter-lick interval: <0.25 s). Time 0 of a lick bout was the timing of the first licking of an individual bout. "Reward delivery" was the water ejection timing from the reward port and was also used in the encoding model analysis (Supplementary Fig. 1).

**Noise correlation**. We examined the correlations of the event-to-event CS response magnitude variations between the simultaneously recorded cells (noise correlation). For calculating the noise correlation, we used data in the period ranging from −0.2 s to 0.1 s following the onset timings of lever-pulls reaching the higher threshold from the resting position. Moreover, the levers were required in resting positions for >1 s before their pull. Considering similar lever trajectories across multiple pulls, we could minimize the variation in CS activities related to those in the lever trajectory in cells encoding the lever positions. We summed the dCSs of individual cells during individual lever pulls and subtracted the predicted number of spikes from the corresponding encoding model. We analyzed cells with ≥10 lever-pull events. Consequently, we calculated the Pearson's correlation coefficients between the adjusted numbers of the two cells as a noise correlation. We compared the noise correlations of the pairs of cells within and across functional types, which were mediolaterally separated (<100 μm), to control for the effect of distance.

We performed data analysis, except with Suite2P, using a custom program written in MATLAB (R2022a, MathWorks).

**Statistics and reproducibility**. Data were collected from nine imaging regions in eight mice. We applied the encoding model analysis on 517 cells. In 352 cells out of them, the activities were predicted by the constructed encoding model significantly. For 352 cells, we examined their response properties, functional arrangement, and noise correlations. All tests were two-tailed, and the significance level was set at $p = 0.05$.

**Reporting summary**. Further information on research design is available in the Nature Portfolio Reporting Summary linked to this article.

## Data availability

The datasets generated during and/or analyzed during the current study are available from the corresponding author upon reasonable request. The numerical source data for the figures were provided as Supplementary Data 1.

## Code availability

The analysis code used in the current study is available from the corresponding author upon reasonable request.

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

## Acknowledgements
We appreciate Drs. Tadashi Yamazaki, Masanori Matsuzaki, and Mitsuo Kawato for the fruitful discussions. The PCP2-Cre mice were courtesy of Dr. Michisuke Yuzaki. We also thank Naomi Yaguchi and Keiko Okazaki for animal care and technical assistance. This research was supported by JSPS KAKENHI (JP19K07801, JP22K07324 to K.I., JP19K06883 to S.M., JP19H05208, JP19H05310, JP19K06882 to M.M., JP18H04012, JP20H05915, JP21H04785 to M.K., JP17H06313, JP22H05161, JP22H00460 to K.K.), a Grant-in-Aid for Brain Mapping by the Integrated Neurotechnologies for Disease Studies (Brain/MINDS) (JP19dm0207079 to S.M., JP19dm0207080 to K.K.), Fundacao para a Ciencia e a Tecnologia (PTDC/MED_NEU/32068/2017 to M.M.), Takeda Science Foundation (to M.M. and K.K.), and the Uehara Memorial Foundation (to K.K.). It was also supported by The Frontier Brain Research Grant from University of Yamanashi. We would like to thank Editage (www.editage.com) for English language editing.

## Author contributions
K.I., N.H., and K.K. conceptualized the study. N.H., S.T., and Y.I. developed the lever-pulling task. N.H. and S.T. performed experiments. K.I., N.H., S.M., and M.M. analyzed the data. K.I., N.H., M.K., and K.K. wrote and revised the manuscript with inputs from all authors. K.I., S.M., M.M., M.K., and K.K. obtained funding. M.K. and K.K. supervised the study.

## Competing interests
The authors declare no competing interests.
