## [Peer Review File · Communications Biology]

REVIEWERS' COMMENTS:

Reviewer #1 (Remarks to the Author):

In this 2P imaging study, the Kitamura lab evaluated the responses of cerebellar Purkinje cells (PC) in a self-initiated reaching paradigm. Focusing on the vermis of lobule V, the Ca²⁺ responses are recorded from PC dendrites and represent activation by the climbing fiber input. A deconvolution approach was used to extract individual Complex Spikes (CSs). An encoding model was employed to develop temporal filters which are used to represent the CSs responses in relation to several behavioral parameters. The authors main conclusion are that the CS responses of a single PC may be modulated by multiple parameters, such a lever movement, licking and reward delivery. Further, the CS responses of neighboring PCs have similar responses, that is there is evidence for spatial grouping, and the CS share correlated noise.

The role of CSs in the cerebellum is of major interest and the 2P approach to examine their properties is appropriate. The authors responded to my original review at Nature Communications. They have undertaken a thorough effort at addressing concerns and issues raised. The manuscript has improved and represents a valuable contribution to the cerebellum literature. This reviewer has no more concerns.

Reviewer #2 (Remarks to the Author):

The authors have responded to previous reviews in a satisfactory manner. The work is well done and reported responsibly and thoroughly. This is a solid and important addition to our understanding of the spatial organization of climbing fiber and coding of sensory, motor, and non-motor variables.

As a minor note, in places where the p-value is not significant, it would be better to give the exact value, i.e. "p=0.3" rather than "p greater than 0.05".

Reviewer #3 (Remarks to the Author):

This is a response to the authors' resubmission. My original review contains information about the major claims of the paper and its relevance for the field.

Overall I am not hugely convinced by the authors response to reviews. Following revision, the data remain interesting but the study does not seem to offer sufficient novel information in isolation.

1) To me it seems a general issue could result from behaviour and performance being quite different between different animals: namely multiplexing within animals may not be the primary cause of the different functional classifications of CS responses.

The authors highlight that two mice performed substantially better than others. Presumably there are further differences between the poorer performers in the way they respond during the task. This suggests that individual mice may have quite a different behavioral repertoires and may be representing different task elements via their CFs. However, CS activities from all mice are grouped together to generate response types. Although 8 functional types are reported, only a few types are typically present within each mouse FOV, especially when grouped within spatially clustered representations (shown in Figure 4). For example, Type 7 seems to be a 'reward kernel' and these are clustered in two mice (Fig 4c,d and Fig 4h) only. There are a few isolated type 7 CSs in other FOVs, but these are not clustered. Presumably there is some noise/variability in the assignment of dCSs to different types so it is unclear the extent to which this could account for highly fractured distributions of functional types. So, in general there may be a smaller number of different behavioural parameters

represented in each mouse, and this may in part be due to the very different levels of task competence (which itself could be unstable). There is a lot of uncertainty here and, all together, this seems to make the main conclusions of the paper rather provisional.

2) The authors are only able to measure the lateral width of two functional clusters – this is disappointing. More detail about the topography of functional signals in cerebellar cortex would be of value to the field.

3) “At the population, the correlation coefficients between the behavioral parameters were 0.09 ± 0.09 (mean \pm standard deviation) as described in Methods.”

Is each mouse considered separately? Otherwise, there may be unique correlations present in the behavior of each animal that are watered down by considering all the data from across mice.

4) In response to reviewer 1, the authors highlight that “In reanalyzed data, the weak modulation for the reward became statistically insignificant.”

This is important to report, but the disappearance of significance in almost half the reanalyzed cells is concerning. What is the reason for this?

5) “We calculated the ratio of the third quartile to the first quartile of the speed of lever-pulls and the duration of lever- holding periods in the individual sessions. Mean \pm s.e.m were 2.4 ± 0.03 for the lever-pull speed and 5.7 ± 0.3 for the duration of a lever-holding period.”

Is this not consistent with a weak correlated-increase in ratios for speed and holding within a session?

First of all, we thank all the reviewers for their constructive comments on the previous version of the manuscript. According to their comments, we have revised the manuscript.

Below are our point-by-point responses to the reviewers' comments and our explanations about how we have changed the manuscript. The reviewers' comments are in bold text, and our responses are in regular text. The part of the main text related to each comment is indicated in red.

Additionally, we changed Figure 1b because we removed a single data that was mistakenly included in the current data set.

Reviewer #2:

As a minor note, in places where the p-value is not significant, it would be better to give the exact value, i.e. “p=0.3” rather than “p greater than 0.05”.

We thank the reviewer for pointing them out. We changed the expression of $p \geq 0.05$ (Lines 109 and 111 in the original manuscript) as follows.

“We did not find the difference between halves of the speeds ($p = 0.11-0.98$, Wilcoxon's rank-sum test). Also, we did not find the difference between halves of the holding durations ($p = 0.21-0.96$, Wilcoxon's rank-sum test) except an imaging site ($p = 0.0002$).”

Additionally, we changed two sentences where the expression of $p < 0.05$ was used to indicate the p-value in single data in L347 and L358 as follows.

“Noise correlations of pairs belonging to identical functional clusters were higher than those belonging to different functional clusters (Fig. 6c, $p < 10^{-5}$, two-sided Wilcoxon's rank-sum test).”

“Red cross: median. $p < 10^{-5}$, Wilcoxon's rank-sum test.”

Reviewer #3:

1) To me it seems a general issue could result from behaviour and performance being quite different between different animals: namely multiplexing within animals may not be the primary cause of the different functional classifications of CS responses.

The authors highlight that two mice performed substantially better than others. Presumably there are further differences between the poorer performers in the way they respond during the task. This suggests that individual mice may have quite a different behavioral repertoires and may be representing different task elements via their CFs. However, CS activities from all mice are grouped together to generate response types. Although 8 functional types are reported, only a few types are typically present within each mouse FOV,

especially when grouped within spatially clustered representations (shown in Figure 4). For example, Type 7 seems to be a ‘reward kernel’ and these are clustered in two mice (Fig 4c,d and Fig 4h) only. There are a few isolated type 7 CSs in other FOVs, but these are not clustered. Presumably there is some noise/variability in the assignment of dCSs to different types so it is unclear the extent to which this could account for highly fractured distributions of functional types. So, in general there may be a smaller number of different behavioural parameters represented in each mouse, and this may in part be due to the very different levels of task competence (which itself could be unstable). There is a lot of uncertainty here and, all together, this seems to make the main conclusions of the paper rather provisional.

As reviewer #2 pointed out, the number of functional types, eight, is not definitive in this study. Because the individual imaging sites cover only a part of lobule V left vermis, we observed several functional clusters in individual sites. We could not examine how many functional types were represented in individual mice. It is possible that the difference of behaviors among mice was reflected in the functional types observed in individual mice. To reveal the functional architecture in lobule V more comprehensively, it is required to record the activities of cells in a sufficiently large area of lobule V and examine their response properties to a wide repertoire of behavior of mice.

We added the paragraph in Discussion as follows.

“The number of functional types reported in this study, eight, is not definitive. Because the individual imaging sites cover only a part of lobule V left vermis, we observed several functional clusters in individual sites. We could not examine how many functional types were represented in individual mice. It is possible that the difference in behaviors among mice was reflected in the functional types observed in individual mice. It is required to record the activities of cells in a sufficiently large area of lobule V and examine their response properties to a wide repertoire of behavior of mice to reveal the functional architecture in lobule V more comprehensively.” (L452–459)

2) The authors are only able to measure the lateral width of two functional clusters – this is disappointing. More detail about the topography of functional signals in cerebellar cortex would be of value to the field.

We agree that more detail about the topography is important for a deeper understanding of information processing in the cerebellum. Unfortunately, the size of imaging sites was insufficient to measure the width of most functional clusters. Only the widths of the two clusters were

measurable, whereas the border of only one side of the other clusters was contained in the imaging site. Instead, we calculated the regression line to examine the signal correlation and mediolateral (ML) distance between cells with $< 200 \mu\text{m}$ of ML distance (Fig. 5f in the new manuscript). The regression line was $y = -0.0023 + 0.5x$, indicating that the signal correlation was close to zero for pairs with approximately $200 \mu\text{m}$ of ML distance. This is consistent with the lateral width of functional clusters. We will reveal the whole picture of the topography of functional signals in a future study. We added the following sentences.

“We measured the lateral width of the two clusters, which appeared to be included in the imaging sites entirely.” (L239–241)

“The regression line for the signal correlation and mediolateral (ML) distance between cells with $< 200 \mu\text{m}$ of ML distance (Fig. 5f) indicated that the response properties of two cells with approximately $200 \mu\text{m}$ of ML distance were dissimilar to each other.” (L260–263)

“The white line and shade in the panel are a regression line for the pairs with $< 200 \mu\text{m}$ of ML distance ($n = 7188$) and its standard error.” (L980–982)

3) “At the population, the correlation coefficients between the behavioral parameters were 0.09 ± 0.09 (mean \pm standard deviation) as described in Methods.”

Is each mouse considered separately? Otherwise, there may be unique correlations present in the behavior of each animal that are watered down by considering all the data from across mice.

Thank you for pointing them out. We followed your advice and corrected the sentences as follows. “Moreover, L2-regularization deals with multicollinearity among explanatory variables, although the behavioral variables were weakly correlated (0.09 ± 0.09 , mean \pm standard deviation for absolute values of correlation coefficients. The mean values ranged from 0.07 to 0.10 across mice).” (L635–636)

4) In response to reviewer 1, the authors highlight that “In reanalyzed data, the weak modulation for the reward became statistically insignificant.”

This is important to report, but the disappearance of significance in almost half the reanalyzed cells is concerning. What is the reason for this?

We examined the significance of kernel modulation for each behavioral parameter in a type-wise manner. In the first manuscript, types 2, 3, 4, and 7 showed significant modulation for the reward

delivery. On the other hand, in the second manuscript, only types 3 and 7 showed significant modulation. Therefore, the significance disappeared in almost half neurons in the reanalyzed data. In the reanalyzed data, the number of neurons whose activities were predicted by the model decreased. This partly causes the increase of the bootstrapped confidence intervals of the kernels, leading to the disappearance of significance in types 2 and 4.

5) “We calculated the ratio of the third quartile to the first quartile of the speed of lever-pulls and the duration of lever- holding periods in the individual sessions. Mean \pm s.e.m were 2.4 ± 0.03 for the lever-pull speed and 5.7 ± 0.3 for the duration of a lever-holding period.” Is this not consistent with a weak correlated-increase in ratios for speed and holding within a session?

We are not sure that we understand the reviewer’s concern correctly. In this analysis, we evaluated the variability of the parameters in individual sessions. We calculated the ratios instead of standard deviation because the frequency distribution of the parameters skewed toward the large value. The ratios were unrelated to the time series of the parameters in sessions.

Per the comments from Reviewer #3, we would ask that you clearly state the limitations of your approach in the Discussion.

We added a paragraph in the Discussion as follows.

“The number of functional types reported in this study, eight, is not definitive. Because the individual imaging sites cover only a part of lobule V left vermis, we observed several functional clusters in individual sites. We could not examine how many functional types were represented in individual mice. It is possible that the difference in behaviors among mice was reflected in the functional types observed in individual mice. It is required to record the activities of cells in a sufficiently large area of lobule V and examine their response properties to a wide repertoire of behavior of mice to reveal the functional architecture in lobule V more comprehensively.” (L452–459)